# Assessment of Diversity of Culturable Marine Yeasts Associated with Corals and Zoanthids in the Gulf of Thailand, South China Sea

**DOI:** 10.3390/microorganisms8040474

**Published:** 2020-03-26

**Authors:** Chutima Kaewkrajay, Thanongsak Chanmethakul, Savitree Limtong

**Affiliations:** 1Department of Microbiology, Faculty of Science, Kasetsart University, Bangkok 10900, Thailand; kchutima@aru.ac.th; 2Division of Microbiology, Faculty of Science and Technology, Phranakhon Si Ayutthaya Rajabhat University, Phranakhon Si Ayutthaya 13000, Thailand; 3Program in Science, Faculty of Science and Technology, Phuket Rajabhat University, Phuket 83000, Thailand; chanmethakul@gmail.com; 4Academy of Science, The Royal Society of Thailand, Bangkok 10300, Thailand

**Keywords:** biodiversity, marine yeast, corals, zoanthids, Thailand

## Abstract

Marine yeasts can occur in a wide range of habitats, including in marine invertebrates, in which they may play important roles; however, investigation of marine yeasts in marine invertebrates is scarce. Therefore, this study aims to explore the diversity of yeasts associated with corals and zoanthids in the Gulf of Thailand. Thirty-three coral and seven zoanthid samples were collected at two sampling sites near Mu and Khram islands. Fifty yeast strains were able to be isolated from 25 of the 40 samples collected. Identification based on sequence analyses of the D1/D2 domain of the large subunit rRNA gene revealed a higher number of strains in the phylum Basidiomycota (68%) than in the phylum Ascomycota. The ascomycetous yeasts comprised nine known species from four genera (*Candida*, *Meyerozyma*, *Kodamaea*, and *Wickerhamomyces*), whereas the basidiomycetous yeasts comprised 10 known species from eight genera (*Vishniacozyma*, *Filobasidium*, *Naganishia*, *Papiliotrema*, *Sterigmatomyces*, *Cystobasidium*, *Rhodotorula*, and *Rhodosporidiobolus*) and one potentially new species. The species with the highest occurrence was *Rhodotorula mucilaginosa*. Using principal coordinate analysis (PCoA) ordination, no marked differences were found in the yeast communities from the two sampling sites. The estimation of the expected richness of species was higher than the actual richness of species observed.

## 1. Introduction

Marine yeasts are yeasts in marine environments that show better growth in seawater than in fresh water [1]. They were first recorded in 1894 by Bernhard Fischer, who isolated yeasts from Atlantic Ocean seawater and identified them as *Torula* sp. and *Mycoderma* sp. [2]. Following this discovery, marine yeasts from different habitats in various localities were then investigated. Known as “facultative marine yeasts”, some marine yeasts originated in terrestrial habitats but, after being washed into the sea, are able to survive in the marine environment. On the other hand, obligate, or indigenous, marine yeasts are unique to marine environments [3]. Marine yeasts may be saprophytic, mutualistic, commensalistic, or parasitic, and can occur in a wide range of habitats including seawater, seaweeds, sea sediments, marine invertebrates, and vertebrates, as well as mangrove ecosystem habitats [4,5,6,7,8].

Corals and zoanthids are marine invertebrates classified into many genera in the subclasses Octocorallia, Hexacorallia, and Ceriantipatharia in the class Anthozoa of the phylum Cnidaria [9,10]. Corals typically live in compact colonies of many identical individual polyps [10]. They secrete a calcium carbonate cup (calicle) around themselves, while zoanthids lack a calcium carbonate skeleton [11]. Corals are commonly found in waters of shallow depth (<30 m) in tropical areas, but coral reefs are also found in deep water and cold water, although in smaller numbers [12,13]. However, zoanthids are commonly found in environments ranging from shallow subtropical and tropical coral reefs to cold seeps in deep seas [14,15].

Corals and zoanthids are known to harbor diverse consortiums of microorganisms, including bacteria, archaea, fungi, viruses, and algae [7,16,17]. Microorganisms associated with corals, including yeasts, are suspected of playing a key role in coral biology by contributing to the nutrition, defense, immunity, and development of corals [10,18]. The associations between bacteria and corals [16,19,20] have received more attention than research on yeasts associated with corals [7] and zoanthids, on which information is scarce worldwide. The few articles have reported investigations of culturable yeasts associated with corals at deep-sea hydrothermal sites in the Mid-Atlantic Ridge, the South Pacific Basin, and the East Pacific Rise (found during oceanographic cruises) [21] and of zoanthids on a Brazilian reef [7]. 

Thailand is a tropical country, rich in biodiversity in plants, animals, and microorganisms (bacteria, yeasts, and filamentous fungi); however, the microbial diversity, especially that of yeasts, has received less attention compared to the diversity of plants and animals. Only a few reports have been published to date on yeast diversity in natural habitats, for example, on the surface and tissue of plant leaves [22,23,24], the soils and peat soils in peat swamp forests [25,26], and water and sediment in a mangrove forest [27,28,29]. Fungi in marine environments, including yeasts, are far less studied than fungi in terrestrial environments, not only in Thailand but also all over the world. Oceans and seas cover more than 12 million km^2^ along the coastline of the Andaman Sea and the Gulf of Thailand [30]; however, only a few investigations have been conducted there on the diversity of microbes, including yeasts [27,28,31,32,33]. Consequently, marine habitats remain one of the habitats in Thailand still to be explored for their microbial diversity. Therefore, this research aims to explore culturable yeast diversity associated with marine invertebrates, that is, corals and zoanthids, in the western part of the South China Sea in the Gulf of Thailand near Mu and Khram islands, Chonburi province, Thailand.

## 2. Materials and Methods 

### 2.1. Collection and Identification of Samples

Coral and zoanthid samples were collected by hand while scuba diving at two sites in the western part of the South China Sea in the Gulf of Thailand, Chonburi province, Thailand. The first sampling site (S1) was at 12°37’9” N, 100°54’25” E, located 622 m off Mu Island (Figure 1). The samples from this site were collected at depths of 15–20 m in March 2016, July 2016, and January 2017 (Table 1). The second sampling site (S2) was at 12°42’19” N, 100°48’27” E, located 274 m off Khram Island (Figure 1). From the latter site, samples were collected at a depth of 3 m in March 2016, July 2016, and February 2017 (Table 1). Each collected sample was put in a clean plastic bag, sealed, and immediately placed in an icebox for transportation to the laboratory. Photographs of hard corals, soft corals, and zoanthids were taken to assist in identification. This was carried out based on morphological data (e.g., structures, tentacles, oral disk/polyp diameter, polyp form) using the methods of Veron [34], Fabricius and Alderslade [35], and Reimer [36]. 

The temperature and salinity of seawater at each sampling site were measured with a sound velocity probe (ATLAS 23120M, Bremen, Germany), while a pH meter (Mettler Toledo, Columbus, Ohio, USA) was used for the pH measurement. 

### 2.2. Marine Yeast Isolation

Upon arrival in the laboratory, the yeasts were immediately isolated. Each sample was rinsed twice with sterile 0.85% sodium chloride (NaCl), then cut into five small pieces (approximately 2.5 cm^3^) and ground with 5 mL of sterile 0.85% NaCl in a sterile blender. After blending, isolation of the yeast was performed using two dilution plate techniques. In the first technique, 0.1 mL of a blended sample was directly spread onto the surface of yeast extract/malt extract (YM) agar prepared with artificial seawater (3 g yeast extract, 3 g malt extract, 5 g peptone, 10 g dextrose, and 15 g agar in 1 L artificial seawater) supplemented with 300 mg L^−1^ penicillin, 300 mg L^−1^ streptomycin, and 250 mg L^−1^ sodium propionate in a Petri dish. The preparation was performed in duplicate. In the second technique, the blended sample (4.9 mL) was added to 50 mL 0.85% NaCl in a 250 mL Erlenmeyer flask and shaken on a rotary shaker at 150 rpm and 20 °C for 1 h, with the precipitate collected by centrifugation at 4000 rpm for 5 min. The precipitate (0.1 mL) was spread onto the surface of YM agar prepared with artificial seawater with the same technique as was used for the first sample. Two replicates were performed. Incubation was carried out at 20 °C for 5–7 days. Yeast colonies with a different morphology from each sample were selected and purified by cross-streaking on YM agar prepared with artificial seawater. All purified yeast strains were preserved in YM broth supplemented with 10% (*v*/*v*) glycerol and maintained at −80 °C. 

### 2.3. Yeast Identification and Phylogenetic Analysis 

The extraction of genomic DNA was performed using the method described by Limtong et al. [37]. If genomic DNA could not be successfully extracted using the cited method, extraction with the solvent mixture of chloroform and isoamyl alcohol (24:1), as described by Ruiz-Barba et al. [38], was used with a slight modification by vortexing for 5 min and centrifuging at 12,000 rpm for 5 min. The amplification of the D1/D2 domain of the large subunit (LSU) rRNA gene and the internal transcribed spacer (ITS) region from the genomic DNA by the polymerase chain reaction (PCR) technique was performed, as described by Limtong et al. [37], using the primers NL1 and NL4 [39] and the primers ITS1 and ITS4 [40], respectively. The PCR product was checked by 1% agarose gel electrophoresis and purified with a TIANquick Midi Purification kit (Tiangen Biotech, Beijing, China), in accordance with the manufacturer’s instructions. The purified products were submitted to First Base Laboratories (Apical Scientific Sdn. Bhd. Company, Selangor, Malaysia) for sequencing of the D1/D2 domain and the ITS region with the same primers used for amplification. 

Yeasts were identified by analysis of the sequence similarity of the D1/D2 domain of the LSU rRNA gene using the BLASTn search program. For identification of yeasts in the phylum Ascomycota, the strains with 0–3 nucleotide substitutions in the D1/D2 domain of the LSU rRNA gene were designated as conspecific, while the strains showing greater than 1% nucleotide substitutions (six nucleotides) were considered to be different species [39]. For the identification of yeasts in the phylum Basidiomycota, the strains showing two or more nucleotide differences were taken to be of different taxa [41]. When the study found more than six nucleotide substitutions in ascomycetous yeasts and two or more nucleotide substitutions in basidiomycetous yeasts, analysis of the ITS sequence similarity was performed: the term “potential new species” was then applied to the strain with no further investigation for description as a novel species in this study. 

Phylogenetic analysis based on the D1/D2 domain of the LSU rRNA gene was performed to verify the identification using sequence similarity analysis. The sequences were aligned with the MUSCLE program in Molecular Evolutionary Genetics Analysis (MEGA) software version 7.0 [42]. A phylogenetic tree was constructed from the evolutionary distance data with Kimura’s two-parameter correction [43] using the maximum-likelihood (ML) algorithm included in MEGA software version 7.0. Confidence levels of the clades were estimated from bootstrap analysis (1000 replicates) [44]. 

### 2.4. Biodiversity Analysis

The similarity of yeast communities in the samples collected at the two sites off Mu and Khram islands was calculated using the classical Jaccard similarity coefficient. The calculation was performed using PAST software version 3.25 [45]. The principal coordinate analysis (PCoA) for the ordination of yeast communities in the 40 samples was based on the Jaccard similarity indices with PAST software version 3.25 [45] employed. The frequency of occurrence (%) was calculated as the number of samples in which a particular species was observed as a proportion of the total number of samples. EstimateS software was used to calculate the species richness on the sampling effort by Chao 1, Jack 1, and Bootstrap estimators with sample-based abundance data (classic EstimateS input) [46]. 

## 3. Results and Discussion

### 3.1. Collection and Identification of Corals and Zoanthids 

The samples were collected from the two sampling sites off Mu (S1) and Khram islands (S2) in 2016 and 2017. The pH, temperature, and salinity of the seawater on each sampling date at each site were measured. The results showed that the pH levels of seawater at both sites were weakly basic (7.94–8.40), the salinity values were 31.2–33.2 practical salinity units (PSUs), and the temperatures were 19.7–30.8 °C (Table 1). 

Our study initially intended to collect only corals, but the morphology of zoanthids was very similar to that of soft corals; therefore, zoanthids were collected along with corals. Samples were collected from each sampling site three times: in March, July, and January or February, with these months being representative of the summer, rainy, and winter seasons of Thailand, respectively. The number of coral and zoanthid samples from sites S1 and S2 were 27 and 13, respectively, with these identified in Table 1. We could not collect the same number of corals and zoanthids at each sampling date in this study as the numbers present were different, which may have been the result of different environmental factors on these sampling dates. The temperatures of seawater at each site on different sampling dates were very different. The temperatures at both sites in March and July 2016 were much higher than those in January and February 2017. Therefore, the temperature might have influenced the number and species of corals and zoanthids present. Changes in climate, including increasing temperatures, changes in precipitation, rising sea surface temperature and rising sea levels have been reported as leading to a reduction in the number of coral reefs: increasing temperature, in particular, has resulted in mass coral bleaching in Mu Ko Surin National Park, Thailand [47]. Corals and coral reefs are extremely sensitive; changes in seawater temperatures can induce reef bleaching [48]. In addition, the different numbers of corals and zoanthids collected at each site may have resulted from the difference in the depths at which the samples were collected. 

### 3.2. Marine Yeast Isolation and Identification 

From the samples collected at site S1, 34 yeast strains were obtained. They consisted of 22 yeast strains from 13 of the 23 coral samples and 12 yeast strains from three of the four zoanthid samples (Table 1). These results indicated that 59.3% of the samples from site S1 contained yeasts. From the samples collected at site S2, 14 yeast strains were obtained from eight of the 10 coral samples, and two yeast strains were obtained from one of the three zoanthid samples. These results indicated that 69.2% of the samples collected from site S2 contained yeasts. 

Identification was undertaken of the 50 yeast strains found in 25 coral and zoanthid samples out of the 40 samples collected from the two sampling sites (Appendix A). The result revealed that 16 strains (32%) belonged to nine known yeast species in four genera of the subphylum Saccharomycotina, phylum Ascomycota, while 34 strains (68%) were identified as 10 known yeast species and one new species belonging to eight genera of the phylum Basidiomycota in two subphyla: Agaricomycotina (4 species, 11 strains) and Pucciniomycotina (7 species, 23 strains) (Table 1 and Table 2, Appendix A). 

The study’s finding was that the number of yeast strains in the phylum Basidiomycota was higher than the number in the phylum Ascomycota, with this finding being aligned with the results of other investigations on yeasts associated with marine animals. For example, more of the marine yeasts isolated from endemic fauna collected from the Mid-Atlantic Ridge, the South Pacific Basin, and the East Pacific Rise during various oceanographic cruises, belonged to phylum Basidiomycota than to phylum Ascomycota [21]. Marine yeasts in sea sponges collected from the North Sea and the Mediterranean were mainly in the order Malasseziales of the phylum Basidiomycota [6]. Furthermore, the most abundant marine yeasts associated with marine animals (phyla Mollusca, Nemertea, Chordata, Cnidaria, Annelida, Echinodermata, Arthropoda, and Platyhelmintes) from Antarctica belonged to the phylum Basidiomycota [49]. However, our study’s result was in contrast with the results of some investigations. Vaca et al. [50] reported that the ascomycetous yeast *Metschnikowia australis* was the predominant yeast organism associated with sea sponges collected in the Antarctic sea. Moreover, a recent investigation reported that of 130 marine yeasts isolated from zoanthids from a coral reef affected by urban sewage discharge, most belonged to the phylum Ascomycota [7]. Therefore, conditions in the marine environment and sources from which yeast are isolated might influence the composition of yeast communities. 

From nine ascomycetous yeast species detected in this study, eight of the nine species (the exception being *Candida spencermartinsiae*), namely, *Candida metapsilosis*, *Candida parapsilosis*, *Candida tropicalis*, *Candida zeylanoides*, *Kodamaea ohmeri*, *Meyerozyma caribbica*, *Meyerozyma guilliermondii,* and *Wickerhamomyces anomalus*, have previously been reported to have been isolated not only from marine environments, such as seawater, sea sediment, sea stars, sea snails, sea sponges and mangrove forest, but also from terrestrial environments, such as tissues and external surfaces of plant leaves, fresh water, and forest soil [21,23,51,52,53,54,55,56,57,58,59,60,61,62,63,64,65]. 

In addition, seven of the 10 basidiomycetous yeast species obtained in this study, namely, *Filobasidium uniguttulatum*, *Naganishia liquefaciens*, *Papiliotrema laurentii*, *Rhodotorula diobovata*, *Rhodotorula mucilaginosa*, *Rhodotorula toruloides*, and *Vishniacozyma victoriae*, were previously reported to have been isolated from marine environments, such as seawater, beach sand, sea sediment, shrimp, sea fish, sea urchins, sea squirts, sea sponges and algae, and from terrestrial environments, such as fresh water and plants [21,51,52,56,58,61,66,67,68,69,70,71,72]. Two basidiomycetous species detected in this study, that is, *Cystobasidium calyptogenae* [73,74] and *Sterigmatomyces halophilus* [75,76], had previously been reported only in marine environments, whereas *Rhodosporidiobolus fluvialis* had never been recorded in a marine environment. 

As detected in this study, ascomycetous yeast species, except for *C. spencermartinsiae*, and basidiomycetous yeast species, except for *Cys. calyptogenae*, *Rh. fluvialis,* and *S. halophilus*, have been found to be present in both marine and terrestrial habitats. Therefore, these yeast species appear to be facultative marine yeasts, which originated in terrestrial habitats and were washed into marine environments where they were able to survive. On the other hand, *C. spencermartinsiae*, *Cys. calyptogenae,* and *S. halophilus* could be obligate or indigenous marine yeast species.

### 3.3. Yeast Diversity

From the 40 samples of corals and zoanthids, 11 samples (27.5%) were found to contain *R. mucilaginosa*. Therefore, this species was the species with the highest occurrence, followed by *P. laurentii,* which was detected in seven samples (17.5%; Table 2). However, *P. laurentii* was detected only in samples collected from site S1 in 2017. The other species were present in one to four samples. Five yeast species, the highest number in one sample, were found in a *Cladiella* sample (W16) collected on 21 January 2017 at site S1. Four yeast species were obtained from *Sinularia* (W1) and three unknown zoanthids (WZ1, WZ2, and SZ1) collected at site S1, and *Sarcophyton* (W21) collected at site S2. The 19 remaining samples contained one to three yeast species. 

From six coral samples of the genus *Cladiella* (W4, W5, W8, W16, W18, and W20) collected at site S1 on 21 January 2017, four samples (W5, W16, W18, and W20) contained yeasts. Among these four samples, the sample W16 contained five yeast species, namely, *Cys. calyptogenae*, *P. laurentii*, *R. diobovata*, *R. mucilaginosa*, and *Rh. fluvialis*, while the other three samples each contained only one yeast species viz. *C. parapsilosis*, *P. laurentii,* and a potential new species closest to *R. toruloides*. It should be noted that different yeast species were detected in these four *Cladiella* samples. Among four coral samples (W3, W7, W17, and W19) of the genus *Klyxum* collected at site S1 on 21 January 2017, three samples (W3, W7, and W17) contained different yeast species. Yeasts were also obtained in two (W6 and W12) of three *Sarcophyton* samples (W6, W9, and W12), each of which contained different yeast species. On the other hand, in the three samples of the genus *Cladiella* (R3, R4, and R6) collected on 30 July 2016 at site S2, one yeast strain, *R. mucilaginosa*, was isolated from each sample. In addition, one unknown coral sample (R5) collected on the same date contained *R. mucilaginosa* together with *C. zeylanoides* and *V. victoriae*. The samples with yeasts from site S2 contained one to four yeast species. Five samples contained one yeast species, whereas two, one, and one samples contained two, three, and four species, respectively. Moreover, the results showed that these yeast species, namely, *C. metapsilosis*, *C. spencermartinsiae*, *Cys. calyptogenae*, *K. ohmeri*, *M. guilliermondii*, *P. laurentii*, *R. diobovata*, *Rh. fluvialis*, *S. halophilus*, *W. anomalus,* and a potential new species closest to *R. toruloides*, were found only in the samples collected at site S1, while *C. zeylanoides*, *F. uniguttulatum*, *R. toruloides,* and *V. victoriae* were found only at site S2. However, *C. parapsilosis*, *C. tropicalis*, *M. caribbica*, *N. liquefaciens,* and *R. mucilaginosa* were found in samples collected at both S1 and S2 sites. 

In the present study, the yeast species collected from the same genus at site S1 on the same date were different. In addition, the yeast species isolated from different genera of the corals collected at the sampling site S2 were similar, and eight of the nine samples contained *R. mucilaginosa*. Therefore, it was very difficult to be able to conclude whether it was the environmental factors or the host species that had more influence on yeast species. A recent study reported that the yeasts collected from zoanthids on the same coral reef were similar, thus showing that the composition of the reef yeast community was influenced more by environmental conditions than the host species [7]. Meron et al. [17] suggested that microbial communities in corals can shift following physiological or environmental change (e.g., in temperature, light intensity, pollution, and salinity). The microbial community in coral mucus has been reported as not very stable due to environmental effects [8]. Conversely, some researchers have reported that the association between corals and their microbial communities was stable and species-specific and often distinct from surrounding microorganisms [19,77]. The fungal community was found to correlate with the host species rather than with differences in the environment [10]. The functions of associations between corals and microorganisms are mostly unknown; however, it was suspected that microorganisms might play an important role in coral health, nutrition, and disease [18]. 

The comparison of the similarity of the yeast communities from the two sites was carried out using the classical Jaccard similarity coefficient. The similarity coefficient value was 0.25, which could mean that both sampling sites shared 25% of the species. The comparison of the yeast communities of all samples was performed by a PCoA plot based on Jaccard similarity indices. The result suggested that no marked differences were found in the similarity of the yeast communities from the two sampling sites (Figure 2). This result was in line with a prior report’s findings that all zoanthid species at Sereia reef and Ponta Verde reef had associated yeast communities that were fairly homogeneous [7]. The estimation of the expected richness of species from the sampling efforts by Chao 1, Jack 1, and bootstrap estimators was higher than the actual richness of the species observed (Figure 3). As species richness estimators perform differently depending on the yeast community structure, increasing the number of rare species results in higher species richness values, as predicted by bootstrap, Jack 1, and Chao 1 estimators. Our results indicated that some yeast species remained unobserved as some yeast species might be difficult to culture, with these results being in line with the findings of other studies [23,24]. 

## 4. Conclusions

Most marine yeasts isolated from corals and zoanthids in this study belonged to the phylum Basidiomycota. The yeast species with the highest occurrence was *R. mucilaginosa*. Many yeast species found in this study had been previously detected in both marine and terrestrial habitats; therefore, these yeasts were designated as facultative marine yeasts. On the other hand, some yeast species (*C. spencermartinsiae*, *Cys. Calyptogenae*, and *S. halophilus*) had been found only in marine habitats, so they were designated obligate, or indigenous, marine yeast species. It could not be concluded from our study’s findings whether differences in the marine environment or in the host species had a greater influence on the yeast species present. Therefore, further investigations of marine yeasts in terms of their ecology and the marine environment should be carried out. 

## Figures and Tables

**Figure 1 microorganisms-08-00474-f001:**
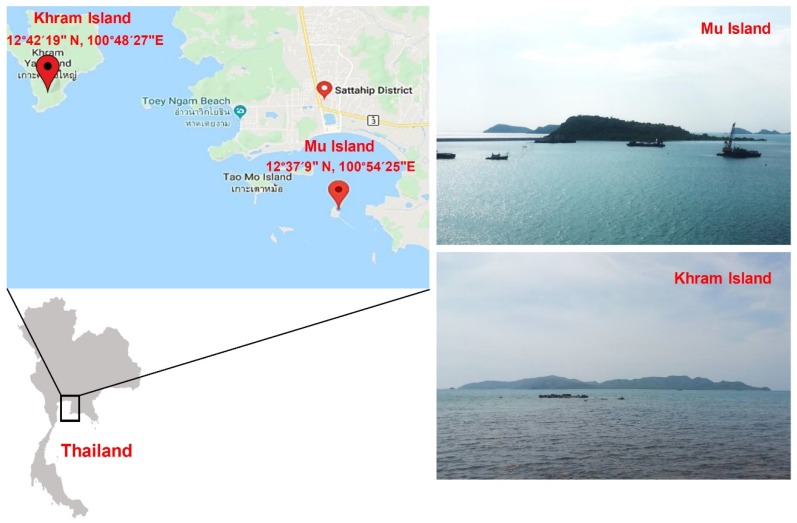
Map of Mu and Khram islands.

**Figure 2 microorganisms-08-00474-f002:**
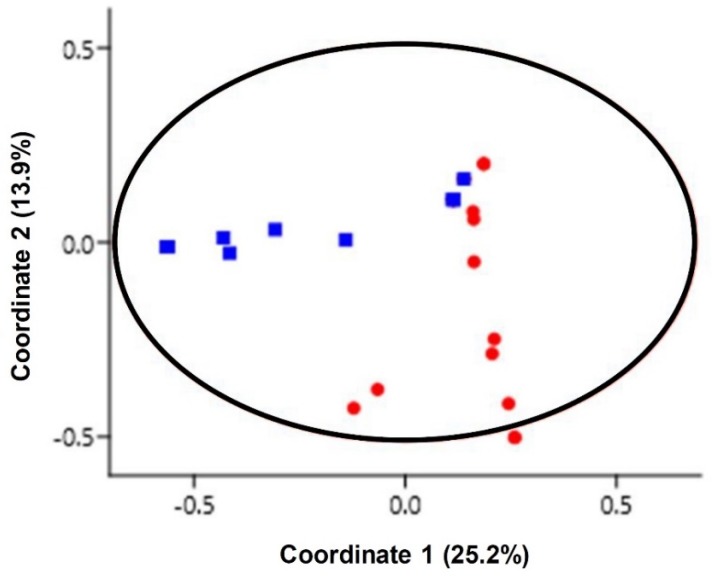
Principle coordinate analysis (PCoA) plots of marine yeast in 40 coral and zoanthid samples of site S1 (filled red circle) and S2 (filled blue square) using the Jaccard similarity coefficient.

**Figure 3 microorganisms-08-00474-f003:**
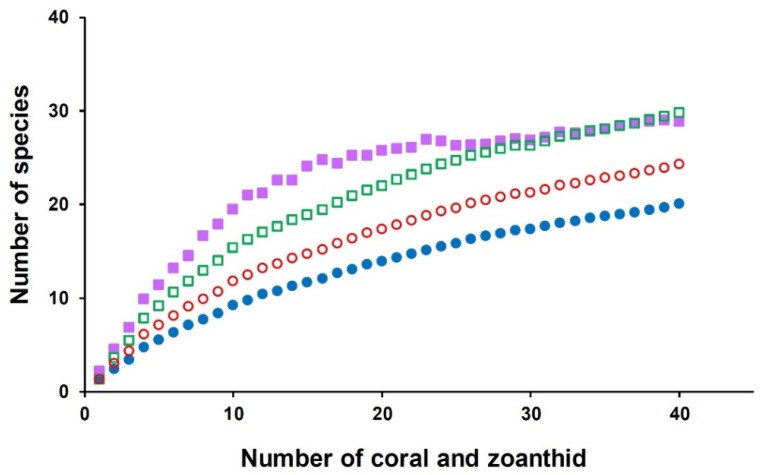
The number of yeast species observed (filled blue circle) from corals and zoanthids, and estimated species richness: Chao 1 (filled purple square), Jack 1 (unfilled square), and bootstrap (unfilled circle) were plotted.

**Table 1 microorganisms-08-00474-t001:** Sampling location, physical properties of seawater in the sampling sites, and yeast isolated from corals and zoanthids.

Collection Date	Physical Property of Seawater			Total Samples	Sample Code	Coral/Zoanthid	No. of Strain	Strain DMKU	GenBank Accession no.	Yeast Species
	pH	Temp. (°C)	Salinity (PSU)							
**Sampling sites S1**										
6 March 2016	8.17	29.4	33.2	2	S1	*Cladiella* sp.	0	ND		
					SZ1	Unknown zoanthid	4	319-1	LC415303	*Meyerozyma guilliermondii*
								319-2	LC415304	*Candida tropicalis*
								C319-1	LC415305	*Wickerhamomyces anomalus*
								C319-2	LC415306	*Candida metapsilosis*
31 July 2016	8.37	30.8	32.7	2	R1	Unknown hard coral	0	ND		
					R2	Unknown hard coral	0	ND		
21 January 2017	8.37	19.7	31.7	23	W1	*Sinularia* sp.	4	J7-1	LC431005	*Papiliotrema laurentii*
								JC7-2	LC431006	*Rhodotorula diobovata*
								JC7-4	LC431007	*Naganishia liquefaciens*
								JC7-5	LC431008	*Candida parapsilosis*
					W2	*Pavona* sp.	1	JC8-1	LC431009	*Papiliotrema laurentii*
					W3	*Klyxum* sp.	1	JC9-1	LC431010	*Cystobasidium calyptogenae*
					W4	*Cladiella* sp.	0	ND		
					W5	*Cladiella* sp.	1	JC14-1	LC431017	*Candida parapsilosis*
					W6	*Sarcophyton* sp.	1	JC15-1	LC431018	*Rhodotorula mucilaginosa*
					W7	*Klyxum* sp.	1	J18-1	LC431020	*Candida spencermartinsiae*
					W8	*Cladiella* sp.	0	ND		
					W9	*Sarcophyton* sp.	0	ND		
					W10	*Acropora* sp.	1	JC22-1	LC497427	*Candida parapsilosis*
					W11	*Sinularia* sp.	0	ND		
					W12	*Sarcophyton* sp.	1	JC24-1	LC511611	*Sterigmatomyces halophilus*
					W13	*Pavona decussata*	2	JC25-1	LC497428	*Papiliotrema laurentii*
								JC25-2	LC497429	*Candida tropicalis*
					W14	*Pocillopora damicornis*	0	ND		
					W15	*Acropora* sp.	0	ND		
					W16	*Cladiella* sp.	5	J33-2	LC497434	*Rhodotorula mucilaginosa*
								J33-3	LC497435	*Papiliotrema laurentii*
								JC33-1	LC497436	*Rhodotorula diobovata*
								JC33-2	LC511612	*Rhodosporidiobolus fluvialis*
								JC33-4	LC511613	*Cystobasidium calyptogenae*
					W17	*Klyxum* sp.	2	JC35-1	LC497442	*Rhodotorula diobovata*
								JC35-2	LC511614	*Cystobasidium calyptogenae*
					W18	*Cladiella* sp.	1	J37-1	LC497437	Potential new species closest to *Rhodotorula toruloides*
					W19	*Klyxum* sp.	0	ND		
					W20	*Cladiella* sp.	1	JC46-1	LC511615	*Papiliotrema laurentii*
					WZ1	Unknown zoanthid	4	J29-5	LC497430	*Meyerozyma guilliermondii*
								JC29-2	LC497431	*Kodamaea ohmeri*
								JC29-6	LC497432	*Papiliotrema laurentii*
								JC29-7	LC497433	*Meyerozyma caribbica*
					WZ2	Unknown zoanthid	4	JC34-1	LC497439	*Papiliotrema laurentii*
								JC34-2	LC497440	*Rhodotorula diobovata*
								JC34-5	LC511616	*Rhodosporidiobolus fluvialis*
								JC34-7	LC497441	*Rhodotorula mucilaginosa*
					WZ3	Unknown zoanthid	0	ND		
**Sampling site S2**										
26 March 2016	7.94	27.1	31.9	4	S2	*Sinularia* sp.	0	ND		
					S3	*Sarcophyton* sp.	1	C322-1	LC415308	*Filobasidium uniguttulatum*
					S4	*Pocillopora damicornis*	2	3222-1	LC415320	*Rhodotorula mucilaginosa*
								3222-2	LC511617	*Naganishia liquefaciens*
					S5	*Sarcophyton* sp.	1	3236-1	LC511618	*Rhodotorula mucilaginosa*
30 July 2016	8.13	28.6	31.9	4	R3	*Cladiella* sp.	1	729-1	LC430111	*Rhodotorula mucilaginosa*
					R4	*Cladiella* sp.	1	7231-1	LC430581	*Rhodotorula mucilaginosa*
					R5	Unknown coral	3	7235-1	LC430589	*Rhodotorula mucilaginosa*
								C7235-3	LC430590	*Candida zeylanoides*
								C7235-4	LC430591	*Vishniacozyma victoriae*
					R6	*Cladiella* sp.	1	7241-1	LC430599	*Rhodotorula mucilaginosa*
18 February 2017	8.40	20.5	31.2	5	W21	*Sarcophyton* sp.	4	FC12-1	LC511619	*Meyerozyma caribbica*
								FC12-2	LC511620	*Rhodotorula mucilaginosa*
								FC12-3	LC511621	*Candida parapsilosis*
								FC12-4	LC511622	*Candida tropicalis*
					W22	*Cladiella* sp.	0	ND		
					WZ4	Unknown zoanthid	0	ND		
					WZ5	Unknown zoanthid	2	F9-1	LC511623	*Rhodotorula mucilaginosa*
								FC9-1	LC511624	*Rhodotorula toruloides*
					WZ6	Unknown zoanthid	0	ND		

ND = Not Detected.

**Table 2 microorganisms-08-00474-t002:** Taxonomic summary of yeast species isolated from 40 coral and zoanthid samples, and their frequency of occurrence.

Phylum	Subphylum	Family	Genus	Species	No. of Strain				Total	Frequency of Occurrence ^a^ (%)
					Mu		Khram			
					2016	2017	2016	2017		
Ascomycota (16 strains)	Saccharomycotina	Debaryomycetaceae	*Meyerozyma*	*M. caribbica*	0	1	0	1	2	5.0
				*M. guilliermondii*	1	1	0	0	2	5.0
			*Candida*	*C. tropicalis*	1	1	0	1	3	7.5
				*C. parapsilosis*	0	3	0	1	4	10.0
				*C. zeylanoides*	0	0	1	0	1	2.5
				*C. spencermartinsiae*	0	1	0	0	1	2.5
				*C. metapsilosis*	1	0	0	0	1	2.5
		Metschnikowiaceae	*Kodamaea*	*K. omeri*	0	1	0	0	1	2.5
		Phaffomycetaceae	*Wickerhamomyces*	*W. anomalus*	1	0	0	0	1	2.5
Basidiomycota (34 strains)	Agaricomycotina	Bulleribasidiaceae	*Vishniacozyma*	*V. victoriae*	0	0	1	0	1	2.5
		Filobasidiaceae	*Filobasidium*	*F. uniguttulatum*	0	0	1	0	1	2.5
			*Naganishia*	*N. liquefaciens*	0	1	1	0	2	5.0
		Rhynchogastremataeae	*Papiliotrema*	*P. laurentii*	0	7	0	0	7	17.5
	Pucciniomycotina	Agaricostilbaceae	*Sterigmatomyces*	*S. halophilus*	0	1	0	0	1	2.5
		Cystobasidiaceae	*Cystobasidium*	*Cys. calyptogenae*	0	3	0	0	3	7.5
		Sporidiobolaceae	*Rhodotorula*	*R. diobovata*	0	4	0	0	4	10.0
				*R. mucilaginosa*	0	3	6	2	11	27.5
				Potential new species closest to *R.toruloides*	0	1	0	0	1	2.5
				*R. toruloides*	0	0	0	1	1	2.5
			*Rhodosporidiobolus*	*Rh. fluvialis*	0	2	0	0	2	5.0
			Total number of yeast strain		4	30	10	6	50	

^a^ Frequency of occurrence (%) = number of samples, where a particular species was observed, as a proportion of the total number of samples.

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
