# Peer review of "Assessment of Diversity of Culturable Marine Yeasts Associated with Corals and Zoanthids in the Gulf of Thailand, South China Sea"

_microorganisms, 2020, doi:10.3390/microorganisms8040474_

Round 1

Reviewer 1 Report

In the present paper the author are presenting the data od the isolation and molecular identification of yeast associated with corals. This is a very nice work that give us a new insight in the yeast world in the marine environments. Will be very nice to have also sequencing data with NGS ....hope for the next article 

Author Response

Reviewer 1

  • English language and style (x) Extensive editing of English language and style required 

Response: The revised MS was English language edited by a Native English speaker, who are not the one that edited the origin MS. Also, English language of the original MS was edited by a native English speaker.

  • In the present paper the author are presenting the data od the isolation and molecular identification of yeast associated with corals. This is a very nice work that give us a new insight in the yeast world in the marine environments. Will be very nice to have also sequencing data with NGS….hope for the next article.

Response: Thank you very much for your suggestion. We hope to study the sequencing data with NGS in our further research.

Reviewer 2 Report

Referees Comments

on the manuscript entitled "Assessment of culturable marine yeasts diversity associated with corals and zoanthids in the Gulf of Thailand, South China Sea” for Microorganisms

The authors studied the diversity of yeast associated with coral and zoanthides in the Gulf of Thailand. The results are new and interesting. The presented manuscript can be published in the Microorganisms, however corrections to the article are required.

  • Page 2, line 65. Please specify that we are talking about filamentous fungi.
  • I did not understand the rationality of the representation of Fig. 1, the text and table 1 contain a detailed description of sampling location.
  • The names of the subsections in the “Materials and Methods” and “Results and Discussions” Sections are the same. It is proposed to remove the names of the subsections from the “Results and Discussion”.
  • The name of the microorganisms in the text must be corrected in italics.
  • Figure 3. Please check if the color for LC431018 is indicated correctly.

Author Response

Reviewers’ comments and suggestions:

Reviewer 2

On the manuscript entitled “Assessment of culturable marine yeasts diversity associated with corals and zoanthids in the Gulf of Thailand, South China Sea” for Micoorganisms

The authors studied the diversity of yeast associated with coral and zoanthids in the Gulf of Thailand. The results are new and interesting. The presented manuscript can be published in the Microorganisms. However corrections to the article are required.

  • Page 2. line 65. Please specify that we are talking about filamentous fungi.

Response: In line 61–65 we talking about yeast “At present, there are only a few reports on yeast diversity in natural habitats such as on the surface and in the tissue of plant leaves [22–24], the soils and peat soils in peat swamp forests [25,26] and water and sediment in mangrove forest [27–29].” But in Line 65, we talking about all fungi that yeast is included.

  • I did not understand the rationality of the representation of Fig. 1, the text and table 1 contain a detailed description of sampling location.

Response: Figure. 1 showed the map of Thailand and sampling sites that we would like to show. So that we moved the detailed of sampling location to Figure 1 and removed the description the sampling location from Table 1.

  • The name of the subsections in the “Materials and methods” and “Results and Discussions” sections are the same. It is proposed to remove the name of the subsections from the “Results and Discussion”

Response: We think the name of the subsections are important. However, according to your suggestion, we changed title of 2.1. in Materials and methods form “Corals and Zoanthids Collection and Identification”  to “Collection and Identification of Samples” inorder to make some different. Page 2, line 75.

  • The name of the microorganisms in the text must be corrected in italics.

Response: We rechecked and italicized name of all microorganisms.

  • Figure 3. Please check if the color for LC431018 is indicated correctly.

Response: We changed color of LC431018 to red color in Supplementary Materials Figure S2.

Reviewer 3 Report

The manuscript describes the isolation an identification of yeast from corals and zoanthids in the Gulf of Thailand

General comments

The methodology used to explore the yeast diversity in corals and zoanthids is not up to date. Nowadays, microbial diversity is estimated using HTS (High Throughput Sequencing). The numerous studies applying HTS to any environment demonstrate that the isolation and identification of 50 yeasts cannot explain the diversity in marine natural ecosystems as corals and zoanthids. I recommend the authors to complement this work with the estimation of fungal diversity using HTS techniques (Example: Alvarez-Yela et al., 2019. Front. Mar. Sci. https://doi.org/10.3389/fmars.2019.00338).

Moreover, the isolation and identification of 50 yeasts from the 25 samples explains poorly the cultivable yeast diversity in corals and zoanthids. The yeast isolation strategy used by the authors prevents the isolation of yeasts representing the ecosystem. The morphology of yeast colonies is usually very similar (creamy, buttery, whitish, etc.), therefore yeast isolation based on differences in morphology is limiting isolation of more yeasts. Important information about the sampling results such as yeast counts from each sample, sampling site or sampling season, etc. are missing in the manuscript. Tables 1 and 2 do not explain yeast diversity but only number of isolates.

The Results and Discussion chapter is a long repetition of the content in the tables and figures.

Figures 2 and 3 do not present new information, they are a representation of the blast result from the sequences used for identification of the isolates. Phylogenies have meaning when new species are described (Keawkrajay and Limtong, 2019. Int. J. Syst. Evol. Microbiol. https://doi.org/10.1099/ijsem.0.003043). In this case Table 1 has the information about the sequence accession number, yeast species and sampling site. The phylogenetic trees in Figures 2 and 3 do not contain sequences of not identified yeasts therefore they have no significance.

Specific comments

The abstract is not very inspiring. I suggest the authors rewrite the first two sentences.

Materials and methods could be shortened replacing the long explanations by literature references already included in the manuscript.

Results and discussion should be shortened. This chapter explains in long detail the content in Tables and Figures. Moreover, the long discussion on yeast species names and where can they be found is not very interesting.

Author Response

Reviewers’ comments and suggestions:

Reviewer 3

  • English language and style (x) English language and style are fine/minor spell check required 

Response: The revised MS was English language edited again by a Native English speaker. Also, English language of the original MS was edited by a native English speaker.

  • The methodology used to explore the yeast diversity in corals and zoanthids is not up to date. Nowadays, microbial diversity is estimated using HTS (High Throughput Sequencing). The numerous studies applying HTS to any environment demonstrate that the isolation and identification of 50 yeasts cannot explain the diversity in marine natural ecosystems as corals and zoanthids. I recommend the authors to complement this work with the estimation of fungal diversity using HTS techniques (Example: Alvarez-Yela et al., 2019. Front. Mar. Sci. https://doi.org/10.3389/fmars.2019.00338)

Response: Thank you for your suggestion. I agree that so far study of microbial diversity by using HTS is becoming more popular. We have experience using both culture-dependent and culture-independent approaches to study yeast diversity in the phylloplane of economic plants in our country (Thailand). However, we are interesting not only to know diversity of cultivable yeast but also to obtain yeast cultures for further utilization in industry (e.g. for microbial oils, indole-3-acetic acid and carotenoids production) and agriculture (e.g. use as biocontrol agent for plant and postharvest diseases). The present study is just a starting of estimation of yeast diversity in marine environment. However, in the future we hope to estimate yeast diversity in marine environment by culture-independent approach using HTS technique.

  • Moreover, the isolation and identification of 50 yeasts from the 25 samples explains poorly the cultivable yeast diversity in corals and zoanthids. The yeast isolation strategy used by the authors prevents the isolation of yeasts representing the ecosystem. The morphology of yeast colonies is usually very similar (creamy, buttery, whitish, etc.), therefore yeast isolation based on differences in morphology is limiting isolation of more yeasts. Important information about the sampling results such as yeast counts from each sample, sampling site or sampling season, etc. are missing in the manuscript. Table 1 and 2 do not explain yeast diversity but only number of isolates.

Response: Thank you for your suggestion. For the isolation methods, at first we tried to use many isolation techniques such as:

  1. Small pieces of corals and zoanthids were directly streaked on the YM agar prepared with artificial seawater.
  2. Small pieces of corals and zoanthids were transferred into 50 mL sterile 0.85% NaCl in Erlenmeyer flask and shaken on a rotary shaker at 150 rpm and 20°C for 1 h. After 1 h, the cell suspension was centrifuged at 4,000 rpm for 5 min. The precipitate was spread onto the surface of YM agar prepared with artificial seawater.
  3. The cell suspension from item 2 was filtered via sterile 0.45 µm cellulose acetate membrane and put this membrane on YM agar prepared with artificial seawater.
  4. Corals and zoanthids were ground with sterile 5 mL of 0.85% NaCl in a sterile blender. 0.1 mL of blended sample was directly spread onto the surface of YM agar prepared with artificial seawater.
  5. Remaining blended sample from item 4 was transferred into 50 mL sterile 0.85% NaCl in Erlenmeyer flask and shaken on a rotary shaker at 150 rpm and 20°C for 1 h. The cell suspension was centrifuged at 4,000 rpm for 5 min and then the precipitate was spread onto the surface of same medium as item 4.

From these isolation methods we found that the method 4 and 5 showed more yeast colonies grow than method 1, 2 and 3. Therefore, we selected method 4 and 5 for further isolation. Additionally, we found a small number of yeast colony from coral and zoanthid samples. Therefore, we did not try to enumerate the number of yeasts by standard method. In our study we used method 4 and 5 for yeast isolation. Unfortunately, we did not count all yeast colonies obtained from each sample, but we selected many yeast colonies that we assumed that they were different base on yeast morphology. When these colonies were purified and identified by molecular technique, we found that many of the collected colony were the same species. Therefore, we selected one isolate as a representative of those isolates that obtained from each sample. For your suggestion, we will apply for our further research.

  • The Results and Discussion chapter is a long repetition of the content in the tables and figures.

     Response: We agree with your suggestion so we moved figures 2 and 3 to Supplementary Materials.

  • Figures 2 and 3 do not present new information, they are a representation of the blast result from the sequences used for identification of the isolates. Phylogenies have meaning when new species are described (Kaewkrajay and Limtong, 2018. Int. J. Syst. Evol. Microbiol. https;//doi.org/10.1099/ijsem.0.003043). In this case Table 1 as the information about the sequence accession number, yeast species and sampling site. The phylogenetic trees in Figures 2 and 3 do not contain sequences of not identified yeasts therefore they have no significance.

     Response: We agree with your suggestion so that we moved figures 2 and 3 to Supplementary Materials.

  • The abstract is not very inspiring. I suggest the authors rewrite the first two sentences.

Response: We rewrote the first two sentences as “Marine yeasts can occur in a wide range of habitats, including in marine invertebrates, in which they may play important roles; however, investigation of marine yeasts in marine invertebrates is scarce.” Page 1, lines 15­–17.

  • Materials and methods could be shortened replacing the long explanations by literature references already included in the manuscript.

Response: We tried to shorten Materials and methods as showed in the revised MS. We remained the parts that have never been reported in the others articles. Page 3, lines 96, Page 7, lines 121–122, 128-129.

  • Results and discussion should be shortened. This chapter explains in long detail the content in Tables and Figures. Moreover, the long discussion on yeast species names and where can they be found is not very interesting.

Response: We tried to shorten “Results and Discussion” as in the revised MS. The other habitats of yeast species detected in this study were reported because we tried to show whether they are facultative or indigenous marine yeast species. However, according to your suggestion we removed the details. Page 8, lines 189, 210–221.

Round 2

Reviewer 3 Report

The authors have taken into account my recommendations and the manuscript has improved considerably. However, the main reasons supporting the poor scientific value of the manuscript: outdated methodology and low number of isolates for estimation of diversity are still the main problems of the manuscript.